# Optimizing energy efficiency in heterogeneous networks: An integrated stochastic geometry approach with novel sleep mode strategies and QoS framework

**Amna Shabbir**[1,2], **Safdar Rizvi**[3], **Muhammad Mansoor Alam**[2], **Faizan Shirazi**[1], **Mazliham Mohd Su'ud**[4]*

**1** Department of Electronic Engineering, NED University of Engineering & Technology, Karachi, Pakistan, **2** Faculty of Computer and Information, Multimedia University, Cyberjaya, Malaysia, **3** Department of Computer Science, Bahria University, Karachi Campus, Karachi, Pakistan, **4** Malaysian Institute of Information Technology, University Kuala Lumpur, Kuala Lumpur, Malaysia

* mazliham@mmu.edu.my

**Data Availability Statement:** All data and code underlying the results presented in this study are available from OSF: 10.17605/OSF.IO/DPM3A.

## Abstract

The quest for energy efficiency (EE) in multi-tier Heterogeneous Networks (HetNets) is observed within the context of surging high-speed data demands and the rapid proliferation of wireless devices. The analysis of existing literature underscores the need for more comprehensive strategies to realize genuinely energy-efficient HetNets. This research work contributes significantly by employing a systematic methodology, utilizing This model facilitates the assessment of network performance by considering the spatial distribution of network elements. The stochastic nature of the PPP allows for a realistic representation of the random spatial deployment of base stations and users in multi-tier HetNets. Additionally, an analytical framework for Quality of Service (QoS) provision based on D-DOSS simplifies the understanding of user-base station relationships and offers essential performance metrics. Moreover, an optimization problem formulation, considering coverage, energy maximization, and delay minimization constraints, aims to strike a balance between key network attributes. This research not only addresses crucial challenges in creating EE HetNets but also lays a foundation for future advancements in wireless network design, operation, and management, ultimately benefiting network operators and end-users alike amidst the growing demand for high-speed data and the increasing prevalence of wireless devices. The proposed D-DOSS approach not only offers insights for the systematic design and analysis of EE HetNets but also systematically outperforms other state-of-the-art techniques presented. The improvement in energy efficiency systematically ranges from 67% (min side) to 98% (max side), systematically demonstrating the effectiveness of the proposed strategy in achieving higher energy efficiency compared to existing strategies. This systematic research work establishes a strong foundation for the systematic evolution of energy-efficient HetNets. The systematic methodology employed ensures a comprehensive understanding of the complex interplay of network dynamics and user requirements in a multi-tiered environment.

**Funding:** The author(s) received no specific funding for this work.

**Competing interests:** The authors have declared that no competing interests exist.

## 1. Introduction

The explosive growth of wireless networks and the extensive increase in high data rate demands in past few years has triggered the academia and industries to put their research efforts towards ubiquitous technology which can satisfy the requirements of high data throughput in a cost-effective manner. In order to achieve these targets, a general consensus among academia, industries, and governments is that instead of designing a new single technology, a mixture of multiple integrated techniques and technologies has been proposed as 5G cellular networks [1]. However, if it is asked what 5G technology is, the only answer with certainty is: it is the *Fifth* generation of cellular networks. Perhaps, it does not provide much information but latest by 2020, when 5G is likely to be rolled out to meet consumers and business demands [1], we may be able to define 5G networks with some certainty. Nevertheless, the good thing is, we know that what consumers are looking for in 5G cellular networks [2]. 5G networks should provide the connectivity across the entire globe and achieve ubiquitous and seamless connectivity to anybody (person to person), anything (machine-person or machine- machine), anywhere (around the world), anytime (wherever they need) and anyhow (by whatever device/networks/services they want). Fig 1 represents the technology shift towards Energy Efficient (EE) communication networks from recent years [3]. It depicts the chronological evolution of research priorities in wireless networks, highlighting a distinctive shift from 2020 towards energy-efficient 5G. The data underscores a transition from an earlier emphasis on capacity and coverage improvement to a more recent dominant focus on reducing energy consumption. The increasing need for traffic data presents a notable challenge, underscoring the critical importance of energy-efficient networks in the present scenario. Notably, the persistent exponential growth in traffic data requirements adds a layer of complexity to achieving energy-efficient networks, posing a substantial challenge for future advancements in the area.

This intensive debate on 5G network fuels a situation of worldwide competition. In the same context, only European Commission (EC) has launched more than ten projects like EARTH, METIS, TROPIC, iJOIN, and MOTO, to investigate the design and architectural needs of upcoming 5G networks. For the period of the last ten years, the EC's half of the investment was allocated to 4G and 5G wireless technologies. The project Horizon 2020, from the period of 2014 to 2020, is the ever large-scale funding for 5G infrastructure, provided by EU research and innovation programs [4]. However, from recent few years attaining Energy

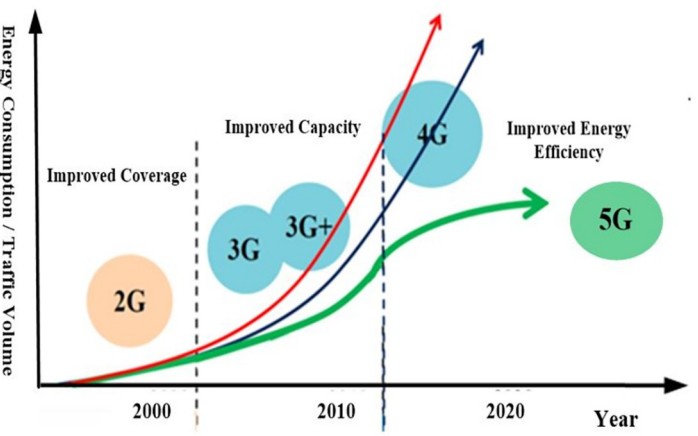

**Fig 1. Shift towards energy efficient networks.**

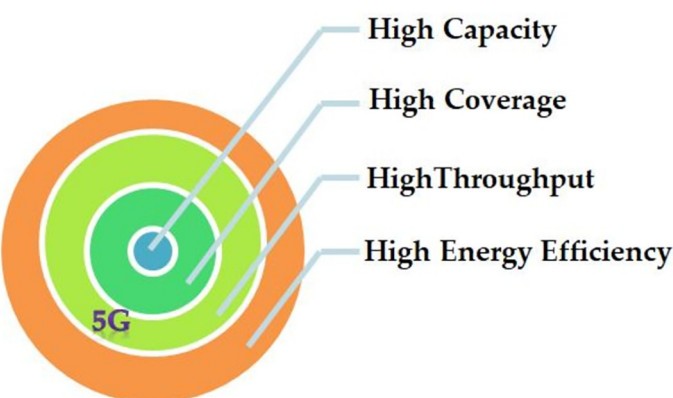

**Fig 2. 5G networks prime objectives.**

Efficiency (EE) along with the provision of high-speed data rate has gained significant attention and become one of the prime objectives of 5G wireless networks. The expectations from 5G are very broad as compare to previous wireless generations (2G, 3G, and 4G etc.) where most of the research efforts have been put towards increasing the data throughput, coverage, fairness and seamless connectivity of the wireless network. It can be inferred from the Fig 2, that 5G cellular network technology is mainly dominated by energy concerns whereas improved network coverage and high network capacity were the prime concerns in previous technologies.

## 2. Heterogeneous Networks (HetNets)-an evolutionary paradigm shift

Heterogeneous Networks (HetNets) has been widely accepted as the paradigm shift in the history of the wireless network. In recent past years, it becomes the prime focus of research academia and network operators due to its potential benefits of fulfilling the need of increased data throughput. Inspired by the possible advantages and attractive features offered by HetNets, the massive development, and deployment of HetNets are gaining momentum in research academia and wireless industries [2, 5–7] Nonetheless, the true deployment of HetNets can potentially achieve high data rate, high coverage, high Quality of Service (QoS) along with the EE utilization of 5G network [8]. Considerable efforts have been dedicated to enhancing the energy efficiency (EE) of current HetNets, resulting in substantial achievements and contributions. However, it is imperative to acknowledge certain significant shortcomings that persist and require thorough investigation. These include an simplified model, unrealistic assumptions, negative impacts on Quality of Service (QoS), the absence of a robust analytical framework for QoS provision, and miscellaneous issues.

One of the key challenges arises from the diversified nature of existing research, making it challenging to establish common points for comparison. For instance, many studies address highlighted issues but often overlook the impact of energy considerations. For example, in [9–11], the authors extensively study average throughput in HetNets Downlink (DL) channel with appropriately biased User Association (UA); yet, energy efficiency analysis is notably absent in their work. Similarly, certain research studies prioritize maximizing energy efficiency using suitable network models [12, 13], but tend to overlook the potential adverse effects on QoS.

In summary, existing research on energy-efficient HetNets reveals several research gaps that warrant rigorous investigation. Upon designing an appropriate Heterogeneous network

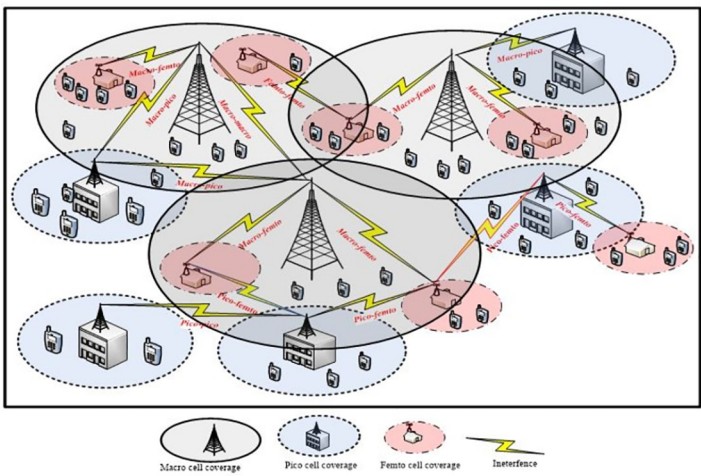

**Fig 3. Multi-tier HetNet architecture.**

topology, the subsequent challenge lies in achieving optimal energy efficiency while maintaining the QoS of the wireless network.

The network architecture of HetNets are likely to be a combination of various networks which differ in terms of its Base Stations (BS) transmit power, sizes and densities. HetNet is the mixture of existing macro BS (homogenous network) under laid by small cell (pico and femto) BSs [14]. This combination of multiple tier is also called multi-tier HetNets, where, macro BSs provide a blanket coverage and the small cells which are introduced to reduce the dead zones or coverage hole and increased capacity at the same time. Fig 3, represents the network architecture of the multi-tier HetNet.

## 3. Related work

With the explosive growth in wireless data traffic demands, HetNets turnout to be a potential solution for improving network throughput, coverage range, and other QoS metric. However, the increasing complexity of HetNets, vast deployment of BSs brings significant challenges related to the energy efficiency of wireless networks. In recent past few years, the research focus remained coverage and data throughput which has been attained through the dense deployment of HetNets while paying less or no attention towards EE aspects. However, it is worth mentioning that the network architecture design, protocols, analysis and computation of non-EE HetNets will be different than that of EE-HetNet. Therefore, to design a unified integrated architecture for EE- HetNet brings a lot of technical challenges and problems to deal with.

Nonetheless, considerable efforts have been done towards improving EE of current HetNets, but achievements and contribution made so far are substantial and have some serious lacking which cannot be ignored. Some of the key shortcoming in the existing HetNets that still need to be investigated which are mentioned below.

Fig 4. reflect the state of the art sleep strategies with respect to switch-off schemes. In HetNet, the Switch-off scheme can be implemented on small cell only or for the overlapping coverage regions. It may seem that switching off macro BS in overlapping region may save more energy as compared to switching off small cell but the aggregate power consumption of small cells may can be larger than macro BS. Hence, more research efforts have been put towards the

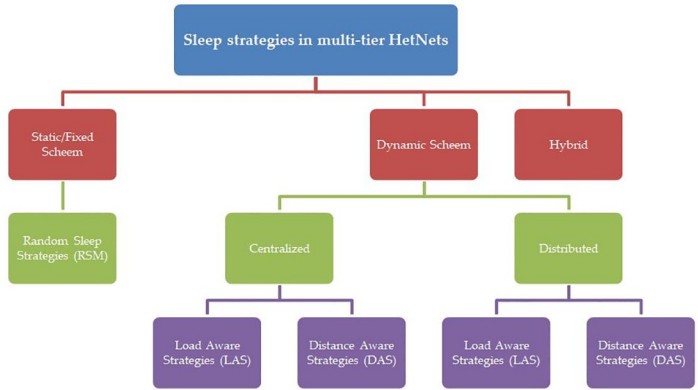

**Fig 4. Switch-off schemes in HetNets.**

design of small cell switch off schemes. Mainly, three approaches are used to implement a switch-off scheme for energy efficiency improvements in HetNets [15–17]. In HetNets, where small cells are expected to enter sleep mode, three broad categories of sleep schemes are employed: static/fixed, dynamic sleep modes, and hybrid. In static/fixed sleep mode schemes, the total number of switches for a specific time period is predetermined, while in dynamic sleep mode schemes, BSs are permitted to switch states as needed based on parameters such as traffic and channel conditions. Although dynamic sleep mode schemes outperform fixed schemes, the increase in the number of switching operations introduces higher computational complexities. Hybrid schemes, a combination of fixed and dynamic sleep mode schemes, are considered the most suitable choice, but the main drawback of hybrid sleep mode lies in the complexity of its implementation. The hybrid scheme requires sophisticated algorithms and mechanisms to seamlessly integrate the fixed and dynamic components, leading to increased computational overhead. Managing the coordination between predetermined schedules and real-time adjustments demands intricate control mechanisms, which can result in higher implementation and maintenance costs.

The approaches may differ in some ways like switching-off algorithm may have fixed or dynamic (centralized, distributed) or hybrid which are further classified into Load Aware Strategies (LAS) and Distance Aware strategies (DAS). The simplest way of switching off small BSs is to turn it on/off independently at some certain probability. Authors in [18–21] have investigated about Random Sleep Mode (RSM) based algorithms. A random sleeping scheme based on times uncorrelated probabilistic models like Bernoulli's process [22–24], the uniform process [25], Poisson process [26, 27], the exponential process [28] etc., have been investigated.

Another technique which has widely been adopted in literatures are based on the intelligent macro BS turning off scheme which are known as DAS algorithms. The Distance-Aware Sleep (DAS) algorithm operates by determining the operational mode (active or sleep) of small cell base stations (BSs) within a HetNet, primarily based on their proximity to the macro BS. The core concept of DAS involves identifying small cell BSs that are closer to the macro BS and strategically turning them off during the sleep mode. This approach is designed to reduce the overall power consumption in HetNets while ensuring the network's efficient operation. The DAS algorithm achieves energy efficiency within HetNets by dynamically adjusting the operational mode of small cell BSs based on their distances from the macro BS. This adaptive strategy takes into consideration both static and dynamic traffic profiles, providing a flexible solution to enhance overall network performance.

Inspired by the work in [29], authors have proposed DAS algorithms for multi-tier HetNet in [30]. The mode of operation (active or sleep) is decided on the basis of the distance of small cell BS from macro BS. For e.g., small cell BSs which are closer to the macro BS are most likely to the turn-off for reducing the total power consumption in HetNets.

Authors in [31], have further proposed DAS based sleeping mode algorithms for static and dynamic data traffic profile. In static data traffic profile, a number of associated users of each nearest small cell BS (within its coverage region) and the distance from the nearest user to each sleeping BS is considered, whereas, in dynamic traffic profile, the distance dynamics are considered in terms of user's mobility direction and velocity. Simulation and results indicate approximately 45% improvement in energy efficiency by using this DAS sleep base strategies. However, due to some obvious reasons like technology limitation, the accurate information about distances between user and BS may not be easily obtained. Furthermore, in a realistic network, the data traffic load distributions at different geographical areas are not uniform. This non-uniform traffic load fluctuation creates an opportunity to minimize energy consumptions by turning off the less utilized small cell BSs. These kinds of strategies are commonly known as LAS. The LAS methodology involves continuous monitoring of small cell BSs traffic load, measured in terms of data rate, with the aim of optimizing energy consumption in HetNets. The methodology employs a threshold to identify underutilized BSs and incorporates time-based criteria, allowing for the strategic turning off of small cell BSs during periods of consistently low traffic. Load-balancing techniques are applied to distribute the load efficiently among different BSs, ensuring overall network performance. Authors in [32] proposed an algorithm based on LAS by turning-off a small BS when its associated traffic load remains below a certain threshold for a certain period of time. It incorporates the load-balancing technique among different BSs while designing the switching-off strategies for small cell BSs.

Results and simulations indicated an energy efficiency improvement up to 33% for medium traffic load and up to 68% for low traffic load. However, in most of the cases, the static data traffic profile is considered for LAS based algorithms. Authors in [36] have proposed a heuristic algorithm for determining which BS to be turned-off based on the number of users associated with the small cell BS. However, this LAS based strategy is not feasible for the realistic network as it requires the knowledge of location and load of all associated users.

Authors in [37] proposed an algorithm by taking user mobility into account. In [37], the HetNet is characterized by a number of users in different BS locations. The state of the network is evolved by using a discrete-time Markov process, whose states are dependent on traffic offloading schemes. When traffic load is low, the macro BS will handle all the users, and all the small cells will go into sleep mode; as the traffic load increases, small cell BS(s) will be turned-on depending upon its user's location and estimated load.

However, this LAS scheme requires the user's localization knowledge. Whereas in realistic networks, the traffic load has variations over time, therefore getting accurate information about user localization will not be an easy task. For example, there is a high chance of fluctuating user's distribution within a day. Authors in [38] have proposed another LAS base algorithm based on only traffic profile statistics with respect to time in the algorithm. Small cell BS is turned off on the basis of a pre-defined fixed timer which is configured manually for a statistical period when users load is very less, like some period in night times. In [39], authors have proposed an adaptive algorithm based on LAS schemes to minimize the time-varying cumulative energy consumption, caused by time-varying static data traffic profile.

The algorithm for switching on\off BSs is derived from complete Channel State Information (CSI) and incomplete CSI cases, respectively. Authors in [40] have proposed a LAS scheme based on optimization of a utility function which is dependent upon data rate, load, and interference. In the proposed model, heuristic and progressive algorithms are adapted for

**Table 1. Switch-off schemes and their limitations.**

| Scheme | Assumptions | Reality | Constraints | QoS | Traffic impact | EE | Comments |
|---|---|---|---|---|---|---|---|
| RSM [18, 22–24] | BSs go to sleep mode according to some pre-defined probability distribution | Traffic densities vary according to data rate requirement of users | Coverage probability | X | x | ↓ | Lack of adaptability to unpredictable conditions like macro BS to small BS distance, BS to MU distance or traffic loads, in realistic networks. |
| | Traffic density is not taken into consideration | | | | | | |
| DAS [29, 31] | BSs can switch their states as much is needed | Unnecessary and frequent change in BS states can create computational complexities | QoS | X | X | ↑ | BS-user distance measurements are very critical |
| | Time delay during switching operations are assumed to be constant or zero | | | | | | |
| LAS [32–35] | Users locations are assumed to be fixed, | Within an active BSs, users can move in across the cell, Outage and blocking probabilities will increase due to QoS degradations | Static traffic profile user localization information | ↗ | X | ↓ for medium traffic | Knowledge of load and location of each MU Perfect localization info will varying from day to day. Static traffic profile |
| | QoS parameters are not taken into account when some BSs are switched off | | | | | ↑ for low traffic | |

turning off unnecessary small cell BSs dynamically. However [39, 40] both assumed the pre-defined data traffic profile which repeats itself periodically. Table 1 provides the comparison of assumption and real scenarios of switch-off schemes used in HetNets.

Another aspect of 5G wirless optimization is the network security and protocol. Authors in [41–45] addresses key challenges in the realm of 5G networks, focusing on optimization, efficiency, and security concerns. Authors in [42] investigate energy efficiency optimization in heterogeneous networks, highlighting issues with traditional handoff strategies and introducing an extreme gradient boosting-based algorithm. The algorithm demonstrates minimal handoff failures, low ping-pong rates, and high success rates, emphasizing the need for additional parameters in optimization to meet diverse user requirements. Future work is suggested to include privacy and security modules for enhanced confidentiality and integrity during handoffs [42].

In the context of 5G networks, [44], analyzes existing intelligent cell selection protocols for neglecting security and privacy. Introducing a protocol based on artificial neural networks and fuzzy logic for target cell prediction during handovers, it achieves high success rates with reduced latencies and communication overheads. Despite the improvements, the study acknowledges the need for further assessment using additional performance metrics in future research [44].

The IoT data safeguarding in 5G is addressed in [43], by introducing a novel security protocol. The proposed protocol is considered suitable for IoT devices as it ensure device authenticity and secure communication with low resource requirements,. However, the paper recognizes the need for a more detailed protocol comparison and exploration of potential vulnerabilities, encouraging real-world testing for a comprehensive evaluation.

Focusing on critical challenges in 5G heterogeneous networks, authors in [45] analyzes legacy handover strategies. Introducing an algorithm utilizing artificial neural networks and fuzzy logic for target cell selection, the paper achieves significant reductions in handover latency and packet losses, aligning with 5G requirements. Future research is suggested to explore the algorithm's security and performance using additional metrics.

The security and privacy concerns in 5G networks are investigated in [41] by introducing a hash signature-based AKA protocol. Security evaluation reveals its robustness against various 5G attacks, with low computation and communication overheads. Considered secure and lightweight, the protocol is proposed for deployment in 5G authentication scenarios. Future work is recommended to formally verify these security features, ensuring their effectiveness. Together, these studies provide valuable perspectives contributing to the continual development of 5G networks, tackling issues related to optimization, efficiency, and security challenges.

## 4. Problem statement

In the past few years, based on the observations presented in Table 1. we see that extensive research has been carried out in the field of HetNets and its energy efficiency improvements [8, 16–18, 31, 32, 34, 46–58]. A comprehensive literature review in the area of 5G-networks reveals that existing network models and algorithms aiming at providing energy solutions are still lacking in the precise solution of EE HetNets. Moreover, another aspect which has been ignored in existing literature is the impact of EE strategies on the architectural design of HetNets and its related QoS parameters. The achievements and contributions made have been substantial and has some serious shortcomings which cannot be ignored. Based on these key aspects, the specific research problems addressed in this dissertation are highlighted as under.,

i) Existing EE models are a generally **simplified model.** Most of the existing EE models have remained focused on single tier with a uniform distribution of BSs locations without considering the effect of multi-tier network.

ii) In the literature where EE aspect has to investigate also have some serious limitations like the use of **unrealistic assumptions** with certain conditions which are too rigid are adopted in their studies.

iii) EE algorithms based on sleep mode strategies, the **negative impact on QoS** (when BSs are turned into sleep mode) has been ignored in various research the publications [16, 59, 60].

iv) Another important factor which has been not investigated in the current literature is the lack of availability of **mathematical framework for QoS provision** or certain key performance metrics for EE HetNets. Therefore there is need to formulate a mathematical framework which not only reflects the impact of EE on HetNet but also computationally efficient and tractable as well.

## 5. Paper contribution

This research makes a substantial contribution to the field through the introduction of the D-DOSS method, a meticulously designed approach specifically tailored for multi-tier HetNets. The D-DOSS method strategically deactivates small cell BS with a keen consideration of various critical factors, such as the random spatial network model, appropriate UA, user density variation, and dynamic traffic conditions. This innovative strategy not only resolves challenges related to low computational complexity but also establishes effective solutions for enhancing EE in HetNets while adhering to desired QoS constraints. In this perspective, the key objectives of this research work are as follow,

**i) Comprehensive EE algorithm with realistic models:**
One of the primary objectives of this research is to develop a comprehensive EE algorithm that incorporates fundamental and realistic models. This algorithm is designed to

intricately address the nuances of Energy Efficiency, ensuring an accurate representation of infrastructure heterogeneity and randomness within a real network. By doing so, the research aims to provide a robust foundation for understanding and optimizing EE in the context of HetNets.

**ii) Optimal algorithms for energy efficiency maximization:**
This research endeavors to formulate optimal algorithms that specifically target the maximization of Energy Efficiency in HetNets. These algorithms take into account various factors, including appropriate UA criteria, variations in traffic load, and the non-homogeneous distribution of BS or user locations. By focusing on optimization, the research aims to provide practical and effective solutions for achieving the highest levels of EE in diverse and complex HetNet environments.

**iii) Analytical framework for QoS provision:**
The development of a corresponding analytical framework is a key objective, specifically tailored for addressing QoS provision in EE HetNets. This framework analyzes key performance metrics such as coverage probability, average data throughput, and outage probabilities. By doing so, the research aims to contribute valuable insights into the interplay between EE and QoS parameters, offering a holistic understanding of network performance in HetNets.

**iv) Computationally effective and optimized analytical framework:**
The research further aims to create a computationally effective and optimized analytical framework for EE HetNets, based on stochastic geometry. This framework is designed to provide simple and tractable solutions for analyzing critical aspects of network performance, including energy efficiency, average achievable data rate, and coverage probability.

In summary, this research significantly contributes to the field by presenting the D-DOSS method and addressing key objectives through a comprehensive exploration of Energy Efficiency and QoS parameters in HetNets. The proposed methodologies and frameworks provide valuable insights for the design, optimization, and analysis of HetNets, paving the way for advancements in the field of wireless communications.

## 6. Mathematical preliminaries

A rich branch of applied probability is known as *Stochastic Geometry* [61] which allows the study of random variables on single or higher dimension planes. Stochastic geometry is basically related to the theory of point process. Initially, it was developed for application in astronomy, material, and biological sciences. In recent years, use of stochastic geometry in the context of wireless cellular network and image processing are getting momentum. In the past ten years, researchers have adopted the use of stochastic geometry for cellular networks, cognitive radios and many other types of wireless communications [62–64]. The use of stochastic geometry for the design and analysis of HetNets has become a strong mathematical tool [63–65]. The proposed model is also based on stochastic geometry with Delaunay triangulation and Poisson Voronoi tessellations for hierarchal network optimization and for mobility management in proposed network model. Stochastic geometry is extensively employed as a mathematical model for the design and analysis of HetNet due to its ability to accurately capture the randomness and unpredictability inherent in the deployment of both transmitter and receiver nodes [65]. In order to maintain a fair comparative analysis of different network scenarios, stochastic geometry serves as a crucial foundation [66]. Unlike deterministic models, it allows for a more realistic representation of the spatial distribution of network elements, facilitating the

derivation of semi-closed and closed-form expressions for key performance indicators like Signal-to-Interference Ratio (SIR), channel capacity, data throughput, and coverage probabilities [67]. Given its widespread use, stochastic geometry ensures that the analysis of HetNets remains robust and applicable to real-world conditions, providing insights into the performance of 5G networks in diverse and dynamic environments [68].

## 6.1 Point Process (PP)

A PP is a type of random process, which is used to describe the collection of distributed points. All points are localized in geographical space and time. It is one of the most powerful tools in statistics for design and analysis of random spatial data points in wireless communication as well as in diverse applications such as medical, economics and astronomy. In telecommunication networks [69], it usually describes the statistical patter produced by a randomly co-located point in 2D space $\mathbb{R}^2$. In order to characterize a PP, a usual approach is to define as a measurable mapping $\Phi$ from a completely separable space $\mathbb{R}^d (d \geq 1)$, which is loacally finite and takes only positive-integer values from integer set $Z^+$.

A PP has several classes, such as simple PP, Stationary PP (SPP), non-stationary PP and Poisson PP (PPP). In this research work, we are going to use simple pp, SPP and PPP which will be defined below.

**6.1.1 Simple and stationary point process.** A PP is said to be simple if a multiplicity of any point is at most one [62]. It means that no two points can take the same place on Euclidean space. In addition, a PP can be a stationary PP or non-stationary PP. Stationary holds iff PP remains invariant by its translation.

**6.1.2 Poisson Point Process (PPP).** In stochastic geometry, the most simple and ubiquitous example of a point process is PPP [63]. A PPP can be defined as a simple PP iff;

- all the disjoint variables $A_1, A_2, \ldots, A_n$ of Euclidean space $\mathbb{R}^d$ are indipendent and

- For set $A \in \mathbb{R}^d$, all random variables $\Phi(A)$ follows poisson distribution provided that the expected disjoint points falling within a region $A$ is the measure of region $A$.

- The term 'measure' is related to the area of $A$. The 'measure' is defined as the sum of two measures which are actually the union of the regions which do not intersect each other.

  Mathematically, the probability distribution of isolated point $N(A)$ can be define as,

$$\mathsf{P}(\mathrm{N}(A) = \mathrm{a}) = \frac{(\eta(A))^a e^{-\eta(A)}}{a!} \tag{1}$$

Where $\eta(A)$ represents the measure of region $A$.

In addition to this, in our research work, we have considered a PPP as homogenous PPP, which means that the density of the points will remains constant throughout the Euclidean space.

## 7. System model

A multi-tier network based on [68] is considered. In a heterogeneous network, each tier is modeled using stochastic geometry. All BSs of each tier are spatially distributed according to homogeneous Poison point process (PPP) $\emptyset_i$ with $\lambda_i$ being their densities and $P_i$ as their transmitting power in Euclidean Space $\mathbb{R}^2$.

We have modeled a multi-tier HetNet [70], as shown in Fig 5, through stochastic geometry, where each tier is classified by its type as in. All the BS components represent a particular class which may differ in terms of spatial density, transmit power and supported average data

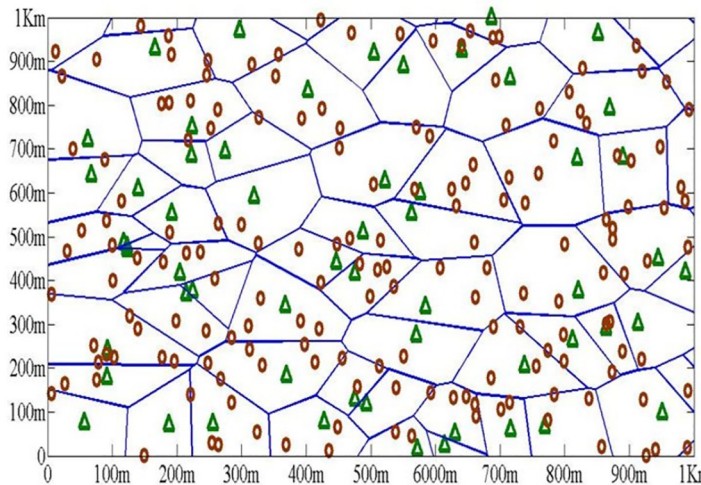

**Fig 5. Multi-tier HetNet with BSs distributed according to PPP with** $⟦(λ)⟧\_femto = 10λ\_macro$**).** Femto (brown) and Macro (green).

throughput. It is further assumed that all BSs and users of $i_{th}$ tier are placed spatially according to some PP $\varphi_i$ of density $\lambda_i$ in Euclidean space $\mathbb{R}$ with power transmitted as $P_i$. All mobile users are also distributed through PPP $\emptyset_m$ with densities being $\lambda_{mu}$. It consists of three tiers (macro-pico-femto), where BSs are located on a VT plot With femto BS density ten times higher than macro BS density($\lambda_{femto} = 10\lambda_{macro}$). The density of pico BSs are kept constant. The threshold value of SINR for successful transmission in each tier is $\beta_{th}$. Without loss of generality we have conduct analysis on a randomly selected mobile user located at origin. That means, any user can connect to its nearest BS in $i_{th}$ tier, as long as its measure SINR in greater than $\beta_{th}$. Therefore, each particular tier can uniquely be identified as a tuple {$P_i$, $\lambda_i$, $\beta_{th}$}.

## 8. Signal-to-Interference and Noise Ratio (SINR) model

The average received power at the typical user from $i_{th}$ BSs located at some point $x_i$ is

$$s_i P_i h_{x_i} \| x \|^{-\propto} \tag{2}$$

Where $z_i \in [0,1]$, represents the power fraction corresponds to the sleep mode describe in later Section 10.1. The expression for SINR of a mobile user can be written as,

$$SINR_{x_i} = \frac{s_i P_i h_{x_i} \| x_i \|^{-\propto}}{I + \sigma^2} \tag{3}$$

Where I is the interference received by the user $x_i$ and $\sigma^2$ is the Additive white Gaussian Noise (AWGN) power. The resulting interference can be represented as

$$I = \sum_{j=1}^{M} \sum_{x \in \emptyset_j \setminus x_i} s_j P_j h_x \| x \|^{-\propto} \tag{4}$$

The resulting SINR can be calculated as

$$SINR_{x_i} = \frac{P_i h_{x_i} \| x_i \|^{-\alpha}}{\sum_{j=1}^{M} \sum_{x \in \Phi_i \setminus x_i} s_j P_j h_x \| x \|^{-\propto} + \sigma^2} \tag{5}$$

**Table 2. Small BS power consumption.**

| Hardware Components | Power consumption (W) [71] | Power consumption with sleep mode (W) [50] |
|---|---|---|
| Microprocessor | 1.7 | 0.5 |
| Microprocessor's associated memory | 0.5 | |
| Backhaul circuit | 0.5 | |
| FPGA | 2.0 | 0.5 |
| FPGA's associated memory | 0.5 | |
| Other related hardware functions | 1.5 | 1.5 |
| RF Transmitter | 1 | 0 |
| RF receiver | 0.5 | 0 |
| RF Power Amplifier | 2 | 0 |
| Total power consumption | 10.2 | 2.5 |

## 9. Power consumption model

In any wireless communication network, the total power consumption can be expressed as

$$P_T = P_{Fixed} + \gamma.P_{tx} \tag{6}$$

$P_T$ and $P_{tx}$ are the total average power consumed and transmitted power by BSs respectively. $P_{Fixed}$ is the fixed power includes miscellaneous power consumptions due to signal processing, cooling of site etc. $\gamma$ is the scaling factor of different radiated power losses likes feeder losses. In designing sleep mode enable networks; hardware design of small cell network is very important because of timely utilization of switching of certain hardware components in low data requirement conditions. In this research paper power consumption model based on [59]is considered. The small cell BSS consist of three integrated blocks namely, microprocessor, Field programmable gate arrays (FPGA) and Power amplifier (PA) respectively. The typical power consumption by small cell BS are provided in Table 2. The total power consumption (in watts) model of small cells based on this hardware corresponds to

$$P_{Total} = P_T + P_{\mu p} + P_{FPGA} + P_{PA} \tag{7}$$

Where $P_T$, $P_{\mu p}$, $P_{FPGA}$ and $P_{PA}$ are power consumptions by the transmitter, microprocessor, FPGA and power amplifier respectively. We have modeled the sleep mode as Bernoulli's trail under RSM strategy where each small cell BS remain active with probability as $q$ and sleeps (switched-off) with $1 - q$ probability. Independent of all other BSs. Therefore, for RSM the power consumption mode will be

$$P_{RSM} = \lambda_{sc}(P_{Fixed} + \gamma.P_{tx})*q + \lambda_{sc}*P_{RSM}*(1 - q) \tag{8}$$

During low data traffic conditions switching off RF front-end elements can save around 40% of power consumption as front-end elements contribute towards largest power consuming can be decreased 75% approximately by reducing consumed power to 2.5W. In addition, switching off these elements takes a few hundred milliseconds to wake up and have no disruptions in small cells operations, as tested in [72].

## 10. Small cell BS activity modes under D-DOSS

In this research work, we adopt power saving modes ordered by 'depth'. Power savings will vary according to the depth which means deeper modes will ensure larger power saving but it takes additional wake up time for small cell BSs.

The study adopts three BS sleep modes [72]:

In **Active Mode** small cells operate at maximum power consumption, ensuring immediate responsiveness. This mode sacrifices energy efficiency for real-time operation.

**Stand-by Mode** involves a light sleep state. Small cells in this mode experience reduced power consumption as components like the RF front-end and heating elements are turned off. The wake-up time from stand-by mode is relatively short compared to deeper sleep modes, striking a balance between energy savings and responsiveness.

**Sleep Mode** represents the most energy-efficient state where small cells consume no power, being entirely powered off. However, transitioning from this mode to an active state incurs a longer wake-up time. This trade-off prioritizes maximum energy efficiency over immediate responsiveness.

The wake-up time is a critical factor, representing the duration for small cells to transition from low-power states to active operation. The choice of power-saving mode involves a trade-off between energy efficiency and the responsiveness of the small cells.

The wake-up time values in the research were obtained from [72], which includes the impact of various wake-up times on the utilization of sleep modes, particularly in the context of small BSs. The values presented in the Table 3 for small cell BSs took into account factors such as network conditions, traffic load, and specific power-saving modes, contributing to an analysis of the variability in wake-up times.

Power utilized in different sleep modes is expressed in terms of activity level and $P_{sleep}$ activity level defines the probability $p_a$ of a small BS to stay in active mode whereas, $P_{sleep}$ is given in Eq 9 defines portion of active powea r used during assigned mode.

$$P_{sleep} = \frac{P_{Active}}{P_{max}} \tag{9}$$

## 10.1 Working principle of D-DOSS

For a multi-tier HetNet, the proposed D-DOSS method turned the small cell BS off while considering the impact of the random spatial network model, appropriate UA, user density variation, and varying traffic conditions. The proposed strategy provides low computational complexity solutions for EE HetNets under QoS constraint. Using tools from stochastic geometry, simple and tractable frameworks for energy efficiency, average achievable data rate and coverage probability are derived analytically.

We have also formulated the optimal trade-off between average throughput and energy by maximizing energy efficiency under QoS constraints of the probability of coverage and average delay in wake-up time. The proposed D-DOSS approach provides interesting insights for design and analysis of EE HetNets.

The proposed D-DOSS outperforms the other state-f the art techniques presented Table 1 and improves EE up to 67% (min side) to 98% (max side). In order to turn off/on a small cell

**Table 3. Small cell attributes in sleep mode.**

| Sleep mode | Wake-up time (in sec) | $P_{sleep}$ (in watts) | Activity level ($p_a$) (in percentage) |
|---|---|---|---|
| Active | NA | 1 | 1 |
| Stand-by | 0.5 | 0.5 | 50 |
| Sleep | 30 | 0 | 0 |

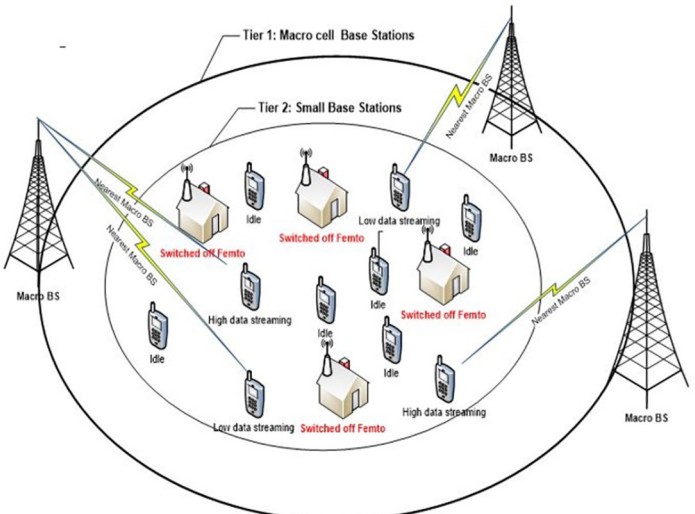

**Fig 6. Network model of proposed D-DOSS.**

BS under D-DOSS sleep scheme is illustrated in Fig 6. None of the small cell BS can be turned off until condition defines in following (a) and (b) both have been satisfied. D-DOSS approach provides interesting insights for design and analysis of EE HetNets.

## 10.2 Sleep mode identification

Instead of RSM and LAS base sleep strategies as discussed earlier in results section, D-DOSS are designed to maximize the EE of HetNets under the QoS constraint. D-DOSS actually works on the basis of throughput requirement and instantaneous SINR levels for switching on/off small cell BSs. A small cell BS can fully operate in an active mode with probability $q_{ON}$ and sleeps with the probability $1 - q_{ON}$, independently. Similarly, $q_{stand\ by}$, $q_{sleep}$ and $q_{off}$ are the probabilities of BS to stay active, stand-by, sleep and switch off modes respectively. Probability assignments have been done by using Bernoulli trail.

Note that all probabilities are independent of each another i-e

$$q_{ON} + q_{stand\ by} + q_{sleep} + q_{off} = 1 \tag{10}$$

A small cell BS can be turned off, iff it fulfills the following conditions

$$\begin{cases} \text{R}_{small\ cell_{average}} < R_{Th} \\ \beta_{macro} > 1 \end{cases} \tag{11}$$

$\beta_{macro} > 1$, it means that a user can connect to at-least one nearest macro BS which provides max instantaneous SINR levels.

The power consumption model will be given as Eq 12

$$P_T = \begin{cases} \lambda_f(P_{Fixed} + \gamma.P_{tx}) & 0 < P_{tx} < P_{max} \\ \lambda_{macro} * (P_{Fixed} + \gamma.P_{tx}) + \lambda_f * P_{sleep} & else\ where \end{cases} \tag{12}$$

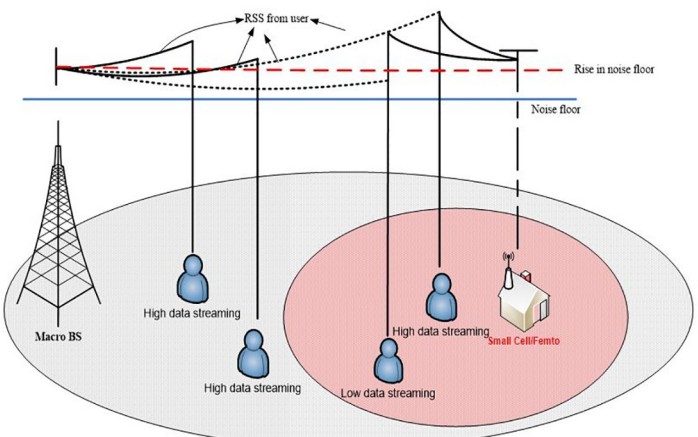

**Fig 7. Illustration of RSS level detection.**

## 10.3 RSS detection

When a mobile user inside the coverage region small cell BS, start voice or data transmission with macro BS, the LP sniffer of small cell detects an increase in received power on UL channel.

The small cell will keep monitoring the average RSS from all mobile users located in its coverage region. The rise in RSS level can easily be detected by the small cell as due high data rate demands of the user(s), there will a clear rise in signal strength on the noise floor when the RSS exceed the threshold value the LP sniffer will send a signal to small cell to start transmission. Fig 7, illustrates the idea behind detection of RSS levels.

## 10.4 Threshold detection

The SINR threshold detection for the proposed model is configured automatically using the path-loss model which uses a power law propagation model, and is the most widely used model in wireless communication. Indeed, this power law model is used by nearly all real wireless networks [73]. In power law propagation model, the signal strength attenuates over a distance. The detection threshold($\Gamma_{Th}$), is calculated by using path-loss to the parent macro BS because the decision that a mobile user will be served by BS is depends upon its path-loss to the BS. This information will be sent to the small cell network and only those users whose RSS levels are higher than the threshold value will be handed over to the nearby small cell. This $\Gamma_{Th}$ will only be used as decesion criteria for shifting users from macro to small cell BSs. However, the mode of sleep will be decided according to criteria provided in Table 3.

## 10.5 User connectivity scheme

An optimal UA scheme for BS cooperation plays a vital role in maintaining acceptable QoS and saving energy of HetNets. In this research paper, an open-access HetNets have been considered, it means that all users are allowed to access any BS from any tier provided that certain UA criterion is satisfied. In addition to that UA is biased towards small cells which provide small coverage and have low transmit powers whiles offloading data traffic from larger BSs (in terms of their sizes and number of users). In this paper Maximum Instantaneous SINR $(\boldsymbol{\beta})_{i_{max}}$ is adopted for UA [74], where any user is allowed to connect to BS which provides the

maximum instantaneous SINR. A typical user can associate with any BS.

$$SINR_k(user)_k > max_{j,j \neq k} SINR_j(user_j)$$

## 11. Performance metrics under D-DOSS

### 11.1 Coverage probability constraint

A mobile user is said to be in the coverage of small cell BS when its RSS exceeds the noise floor as describes in Fig 7. With this understanding coverage probability for D-DOSS is derived for open access mode. Corollary 1 gives the main result for coverage probability.

**Corollary** 1 (with neglected noise effect):

In interference limited environment where self-interference dominates internal noise, the probability of coverage for a typical MU can be expressed as,

$$P_{C_{D-DOSS}} = \frac{\lambda_f q_{ON} P_f^{\frac{2}{\alpha}} \beta_f^{-\frac{2}{\alpha}} + \sum_{j=2}^{K} \lambda_j P_j^{\frac{2}{\alpha}} \beta_j^{-\frac{2}{\alpha}}}{\lambda_f P_f^{\frac{2}{\alpha}} + \sum_{j=2}^{K} \lambda_j P_j^{\frac{2}{\alpha}}} \frac{1}{2\pi csc\left(\frac{2\pi}{\alpha}\right)\alpha^{-1}}; \quad \begin{cases} \beta_f > \beta_{Th} \\ \beta_j > 1 \end{cases} \tag{13}$$

In an interference-limited network the coverage probability is independent of the density of BSs, and only depends on the signal-to-interference $(SIR)^{**}$ threshold ratio. Variables are defined as below,

$P_{C_{D-DOSS}}$: Probability of successful communication in the DL

$\lambda_f$: Density of the interfering primary transmitters.

$q_{ON}$: Probability that the D-DOSS transmitter is on.

$P_f$: Power of the D-DOSS transmitter.

$\beta_f$: Path loss between the D-DOSS transmitter and receiver.

$\alpha$: Path loss exponent.

$K$: Number of interfering D-DOSS transmitters.

$\lambda_i$: Density of interfering D-DOSS transmitters.

$P_i$: Power of interfering D-DOSS transmitters.

$\beta_j$: Path loss between interfering D-DOSS transmitters and the receiver.

$\beta_{Th}$: Threshold for path loss.

**Proof for corollary 1:**

For open access mode, in an interference-limited environment, where thermal noise can be ignored in presence of self-interference, the coverage probability of a typical user can be expressed as [57].

$$P_C(\{\lambda_i\}, \{\beta_i\}, \{P_i\}) = \frac{1}{2\pi csc\left(\frac{2\pi}{\alpha}\right)\alpha^{-1}} \frac{\sum_{i=1}^{K} \lambda_i P_i^{2/\alpha} \beta_i^{-2/\alpha}}{\sum_{i=1}^{K} \lambda_i P_i^{2/\alpha}}; \beta_i > 1 \tag{14}$$

Since a user is said to be in the coverage region when it transmits/receives signal to/from its nearest BSs. With sleep mode schemes, the expression for coverage probability is equivalent to

that of without sleep with a density of active small BSs being $\lambda_f q_{on}$, it leads to Eq 15.

$$P_{C_{D-DOSS}} = \textit{Active Small cell tier BS}_{\textit{sleep mode}} + \textit{Sum of all other tiers}$$

$$P_{C_{D-DOSS}} = \frac{\lambda_f q_{ON} P_f^{2/\alpha} \beta_f^{-2/\alpha} + \sum_{j=2}^{K} \lambda_j P_j^{2/\alpha} \beta_j^{-2/\alpha}}{\lambda_f P_f^{2/\alpha} + \sum_{j=2}^{K} \lambda_j P_j^{2/\alpha}} \frac{1}{2\pi \csc\left(\frac{2\pi}{\alpha}\right)\alpha^{-1}}; \quad \begin{cases} \beta_f > \beta_{Th} \\ \beta_j > 1 \end{cases} \quad (15)$$

## 11.2 Average data rate constraint

In this section average, the achievable data rate is computed when MU is in open access mode, provided that mobile user is in the coverage region of at least one BS. For the sake of simplicity, we have assumed an interference-limited environment with $\sigma^2 = 0$, i-e thermal noise is suppressed by self- interference. The primary result for average achievable data rate is given in Theorem 1.

### Theorem 1

The average achievable data rate in D-DOSS is obtained by,

$$\overline{R_{D-DOSS}} = \frac{\lambda_f q_{ON} P_f^{2/\alpha} \mathcal{A}\left(\alpha, \beta_i, \beta_{min}\right) + \sum_{k=2}^{K} \lambda_k P_k^{2/\alpha} \mathcal{A}\left(\alpha, \beta_k, \beta_{min}\right)}{\lambda_f q_{ON} P_f^{2/\alpha} \beta_f^{-2/\alpha} + \sum_{k=1}^{K} \lambda_k P_k^{2/\alpha} \beta_k^{-2/\alpha})} + \log(1 + \beta_{min}) \quad (16)$$

Where $(\alpha, \beta_i, \beta_{min}) = \int_{\beta_{min}}^{\infty} \max(\beta_i, x)^{-2/}\alpha/1 + x$, and $\beta_{min} = \min\{\beta_1, \beta_2, \beta_3 \ldots \beta_K\}$.

**Where as variables in Eq 16 are defined as**

$R_{D\text{-}DOSS}$: Average achievable rate in the DL

$\mathcal{A}(\alpha, \beta_k, \beta_{min})$: Afunction dependent on the path loss exponent ($\alpha$), path loss between D-DOSS transmitters ($\beta_i$), and minimum path loss ($\beta_{min}$).

$K$: Number of interfering D-DOSS transmitters.

$\lambda_k$: Density of interfering D-DOSS transmitters.

$P_k$: Power of interfering D-DOSS transmitters.

$\beta_k$: Path loss between interfering D-DOSS transmitters and the receiver.

$\beta_f$: Path loss between the D-DOSS transmitter and receiver.

$\beta_{min}$: Minimum path loss.

### Proof

In an HetNet the average achievable data rate by a randomly selected mobile user in open access mode, when user is in coverage is [75],

$$\bar{R} = \log(1 + \beta_{min)} + \frac{\sum_{i=1}^{K} \lambda_i P_i^{2/\alpha} \mathcal{A}\left(\alpha, \beta_i, \beta_{min}\right)}{\sum_{i=1}^{K} \left(\lambda_i P_i^{2/\alpha} \beta_i^{-2/\alpha}\right)}$$

Where

$$\mathcal{A}(\alpha, \beta_i, \beta_{min}) = \int_{\beta_{min}}^{\infty} \frac{\max(\beta_i, x)^{-2/}\alpha}{1 + x} x,$$

and

$$\beta_{min} = \min\{\beta_1, \beta_2, \beta_3 \ldots \beta_K\}. \tag{17}$$

Data transfer will only happen when BSs are in *'active'* mode. In simple words when BSs are in switched off state, no BS will be able to send/receive data to user(s) because its RF module will be switched off that means it will not contribute to interference, in addition, average data rate will be dependent on the number of BSs which are ON ($\lambda_f q_{ON}$). With this approach the achievable average data rate expression for mobile user located at the origin will be,

$$\overline{R_{D-DOSS}} = Active\ Small\ cell\ tier\ BS_{sleep\ mode} + Sum\ of\ all\ other\ tiers$$

Therefore the expression for average achievable data rate by mobile user in D-DOSS can be deduced.

### 11.3 Delay considerations

Now, we consider the constraint for the average delay from sleep mode to wake up mode in small cell BSs. As for saving energy in multiple BSs operations, the controller can turn on/off various BSs simultaneously, since it is very important for BSs to wake-up in a timely manner for providing service to demanding users. Therefore the constraint for an average wake up time for the mobile user can be written as,

$$d_{on}q_{on} + d_{stand\ by}q_{stand\ by} + d_{sleep}q_{sleep} + d_{sw.off}q_{sw.off} \le d_{min} \tag{18}$$

Delay parameters are specified in Table 3, therefore this constraint can further simplify as,

$$q_{on} + 0.5q_{stand\ by} + 30q_{sw.off} \le d_{min}\ (in\ seconds) \tag{19}$$

### 12. Energy efficiency of multi-tier HetNets under D-DOSS method

The EE of HetNet is defined as the ratio of average achievable data rate ($\bar{R}$) and overall power consumption ($\mathcal{P}$) of BSs in $i_{th}$ tier. As discussed previously, the total consumption power in BS is sum the of transmit power, microprocessor power, an FPGA power and PA as mentioned in Table 2. For the sake of simplicity we assumed that transmitting power of all BS is kept to be constant throughout the network which will be the fraction of total power ($\mathcal{P}_i/\mathcal{P}_T$).

*Lemma 1*

EE for D-DOSS can be obtained as,

$$
EE_{D-DOSS} = \left[ \frac{1}{\lambda_f(q_{on} + 0.5q_{stand\ by} + 0.15q_{sleep})\frac{\mathcal{P}_f}{\mathcal{P}_T} + \sum_{j=2}^{K}\left(\frac{\lambda_j \mathcal{P}_j}{\mathcal{P}_T}\right)} \right]
$$
$$
* \left[ \frac{\lambda_f q_{ON} P_f^{2/\alpha} \mathcal{A}(\alpha, \beta_i, \beta_{min}) + \sum_{k=2}^{K} \lambda_k P_k^{2/\alpha} \mathcal{A}(\alpha, \beta_k, \beta_{min})}{\lambda_f q_{ON} P_f^{-2/\alpha} \beta_f^{-2/\alpha} + \sum_{k=1}^{K} \lambda_k P_k^{2/\alpha} \beta_k^{-2/\alpha})} + \log(1 + \beta_{min}) \right] \tag{20}
$$

*Proof of Lemma 1*

The EE of a typical mobile user is the ratio of achievable can be computed conditioned on the mobile user being in its region of coverage to the overall power consumption of network i-e

$$EE = \frac{\Re_{femto,Sleep\ mode} + \Re_{all\ other\ tier}}{\mathcal{P}_{femto,sleep\ mode} + \mathcal{P}_{all\ other\ tiers}}$$

The power per unit area can be calculated as,

$$\mathcal{P}_{femto,sleep\ mode} + \mathcal{P}_{all\ other\ tiers} = \lambda_f(q_{on} + 0.5q_{stand\ by} + 0.3q_{sleep})\frac{\mathcal{P}_f}{\mathcal{P}_T} + \sum_{i=2}^{K} \lambda_i \mathcal{P}_i / \mathcal{P}_T \quad (21)$$

The average data rate per unit area with the assumption that mobile user is in coverage area can be computed as,

$$\Re = E_{\Phi,N_c}\left[\sum_{j=1}^{N_c} \Re_j\right]$$

Therefore the EE can be computed as

$$EE = \left[\frac{\Re}{\lambda_f(q_{on} + 0.5q_{stand\ by} + 0.3q_{sleep})\frac{\mathcal{P}_f}{\mathcal{P}_T} + \sum_{i=2}^{K} \lambda_i \mathcal{P}_i / \mathcal{P}_T}\right]$$

$$= \left[\frac{1}{\lambda_f(q_{on} + 0.5q_{stand\ by} + 0.3q_{sleep})\frac{\mathcal{P}_f}{\mathcal{P}_T} + \sum_{i=2}^{K}\left(\frac{\lambda_i \mathcal{P}_i}{\mathcal{P}_T}\right)}\right] \quad (22)$$

$$* \left[\frac{\lambda_f q_{ON} P_f^{2/\alpha} \mathcal{A}(\alpha, \beta_i, \beta_{min}) + \sum_{k=2}^{K} \lambda_k P_k^{2/\alpha} \mathcal{A}(\alpha, \beta_k, \beta_{min})}{\lambda_f q_{ON} P_f^{-2/\alpha} \beta_f^{-2/\alpha} + \sum_{k=1}^{K} \lambda_k P_k^{2/\alpha} \beta_k^{-2/\alpha})} + \log(1 + \beta_{min})\right]$$

## 13. D-DOSS algorithm description

The detail description of the proposed D-DOSS algorithm is as follows. A small cell will stay in sleep state according to the defined modes and disable all the transmission processing, except the 'Low Power (LP) Sniffer', which is a part of small cell, allows the detection of active mobile users and their average RSS within the coverage area of small cell and parent BS. When the $RSS_{avg}$ level increase from the predefine SINR threshold, the small cell will be activated and starts its pilot transmission and associated processing to mobile users located inside coverage of the small cell. The LP sniffer keeps monitoring $RSS_{avg}$ levels and as soon as this level drops below its threshold value, the LP sniffer sends a signal to small cell to stop its all radio processing and go back to the sleep mode except for the LP sniffer.

### 13.1 Network configuration

The network is divided into $N_c$ equal size squres, where each square represents a cluster region. Howere, the boundaries of clusters only used to identify that from which cluster each BS belongs to and thus does not reflect any changes in UA as all users will still associates with the BS which provide $\beta_{max_i}$.

Fig 8, represents the network designed with clustered dynamically distributed throughout the network. These clustered are connected to the core network through a backhaul network.

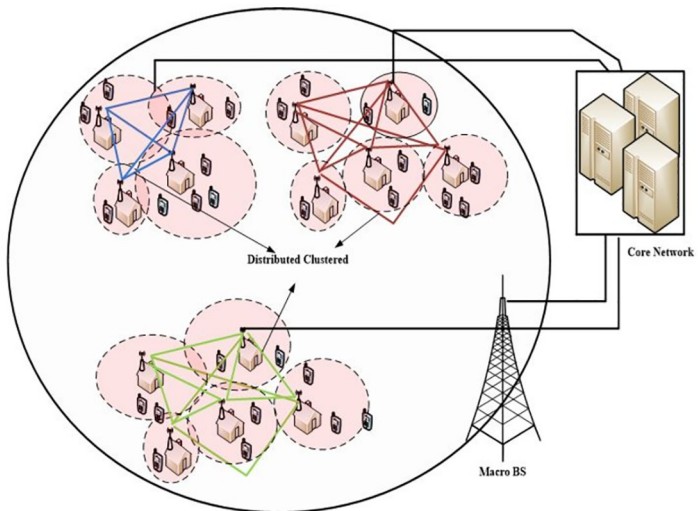

**Fig 8. D-DOSS network design.**

### 13.2 D-DOSS pseudocode

In this section, we provide algorithms for switching off small cell BS using RSM and D-DOSS methods. In RSM strategy BSs are turned off according to some probability criteria and takes pre-defined traffic profile into consideration. However, in D-DOSS approach, small cell BSs are turned off only when it satisfies all the constraint as discussed. The detail description of RSM and D-DOSS algorithms are summarized in Algorithm 1 and Algorithm 2 respectively.

**Algorithm 1: Switching off small cells based on RSM**

```
Step 1:
1-1: Initialize SNR(βᵢ), and assign MU to all BSs
1-2: Randomize βᵢ for all BSs
Step 2:
2-1: Set number of MU
2-2: Set data traffic load (L) for all MU
2-3: a Set optimal threshold value for data traffic load for all MU
based on randomized βᵢ
Step 3:
3-1: Set transmitting power (Pₜ)for all BS
3-2: Assign fraction of Pₜ to those BS whose (L) satisfied threshold
conditions
3-2: switch off remaining BS
3-4: Repeat 2-3 for all BSs
3-5: Update βᵢ, L and recalculate Threshold
Step 4:
4-1: If threshold condition is not satisfied
4-2: Switch on all BS to serve MU
4-3: Repeat 3-4
End process
```

For improved network performance in specifically terms of EE D-DOSS methods can be used. Instead of turning off small BSs randomly, we have adopted opportunity base strategy to serve a maximum number of the active user while reducing energy consumption.

In this mode, users will be served by BSs according to their throughput demands. The fraction of power will be assigned according to Eq 23

$$p_{frac} = (R_i/R_{Th})P_T \tag{23}$$

**Algorithm 2: Switching off small cells based on D-DOSS**

```
Step 1:
1-1: Initialize SNR(βᵢ), and assign MU to all BSs
1-2: Randomize βᵢ for all BSs
Step 2:
Decide the number of BSs in N_ON, N_standby, and N_sleep by rounding N_qON,
N_qstandby, and N_qsleep to an integer value. Where N denotes the number of BS
in small cell zone.
Step 3:
3-1: Set number of mobile user in the coverage area of each small cell
BS
3-2: calculate average data throughput (Rᵢ) of all mobile user
3-3: a Set optimal threshold value for throughput(R_th) for all mobile
user based on randomized βᵢ
Step 4:
4-1: Set transmitting power (Pₜ)for all BS
4-2: Assign fraction of P_frac to those BS whose throughput satisfied
threshold conditions
4-3: Assign Powers to BS according to states i-e P_ON, P_standby, and P_sleep
4-4: Repeat 2-3 for all BSs
4-5: Update βᵢ, Rᵢ and recalculate R_th
Step 5:
5-1: If threshold condition is not satisfied
5-2: Switch on all BS to serve mobile users
5-3: Repeat 3-4
End process
```

The proposed algorithm will not only provide improved EE but also works better in terms of outages which will lead to increase the overall performance of HetNets.

## 14. Numerical results and discussions

This section presents the numerical analysis of the results; we have use simulations to validate the proposed sleep mode strategy. Simulation parameters as given as follow.

### 14.1 Simulation parameters

In this section, our results will be demonstrated through Monte Carlo method with an adequate number of iterations. We use Table 4, as default values unless otherwise stated. Since self-interference dominates noise in typical HetNets, therefore the noise power ($\sigma^2$) is assigned as zero. At each trial of Monte Carlo simulation, the locations of all macro BSs, small cell BS and mobile users are distributed according to PPP in an area of 1 km X 1 km square. Each mobile user is assumed to be connected to at-most one BS which provides the highest channel gain. In addition additive white Gaussian noise is assumed to be zero. Each serving BS selects users on the basis of RSS levels.

### 14.2 Coverage probability constraint

Fig 9, plots the coverage probability versus SINR ratio for different sleep strategies. Coverage probability reduces with the increased number of users serviced by BSs. As seen from the plot the coverage probability of D-DOSS is far better than RSM and LAS strategies. This is because

**Table 4. Simulations parameters.**

| BS distribution | PPP |
|---|---|
| Tier-1 (Macro BSs) Density | 1/500 square meters |
| Tier-2 (Femto BSs) Density | 4/500 square meters |
| The power consumption of Macro BSs | 400W |
| The power consumption of Femto BSs | 40W |
| SIR Threshold of Femto $\beta_f$ | 1.1 |
| SIR threshold for macro $\beta_m$ | 1.3 |
| Path loss exponent | 2 |
| System Bandwidth | 10MHz |
| Path Loss model for macro | L = 128.1+37.6log10(R) (R in km) |
| Path Loss model for small cell | L = 140.7+36.7log10(R) (R in km) |
| MU rate requirement for macro BS | 400kbps |
| MU rate requirement for small cell BS | 400kbps |
| The minimum distance between the macro BS and MU | 35m |
| The minimum distance between the macro BS and small cell BS | 75m |
| The minimum distance between the macro BS and MU | 35m |
| The minimum distance between small cell BS and MU | 10m |
| The minimum distance between two small cell BS | 40m |

in RSM strategy BSs are turned off without considering the status of traffic variations which might increase the chances of decreased coverage due to turning off a small BS during high data traffic demands for example when *SINR* = 6*dB*, the coverage probability is around 0.15% which gradually decreases to 0 when *SIR* levels increases from 18dB. In LAS strategies, the coverage probability is improved to 1% approximately which is due to inclusion of data the traffic variations in LAS strategy. Nonetheless, the coverage probability in LAS strategy when SINR levels raises from 21dB falls to 0. Whereas the D-DOSS scheme outperforms the other two strategies by improving coverage up to 98% this is by marginally lower than no sleep mode.

The reason behind improves coverage is that in D-DOSS strategies small cell BS does not go into sleep mode until all the mobile users transfer to the underlaid macro BS. In addition to that the in RSM and LAS strategies the activity levels are assumed to be 0 and 1 with the equal probability of 0.5, whereas in D-DOSS the activity level is based on Table 3, which will lead towards an increased rate of coverage probability. Fig 10, represents the coverage probability of LAS and D-DOSS strategies with respect to the SINR threshold ratio as only these two strategies use threshold criterion to activate/de-active their BSs. It can be noticed from the plot that D-DOSS has better coverage as compared to LAS.

It is because LAS follows the static traffic profile, which does not account for the unusual increase or decrease in traffic load. For instance, when threshold ratio is less than 0.1 than D-DOSS is marginally better than LAS, whereas for higher ratios like threshold ratio = 0.7, coverage probability for LAS is 70% and with D-DOSS it increases up to 90%. The reason behind this improved coverage is that D-DOSS observes the instantaneous threshold levels instead of the daily static traffic profile like LAS.

## 14.3 Energy efficiency

Fig 11, represents the plot of energy efficiency by in bits/joules. Simulated results show that energy efficiency is considerably low with no sleep mode schemes. A slight improvement can be observed when RSM strategy has been implemented. It can be noticed from the simulated

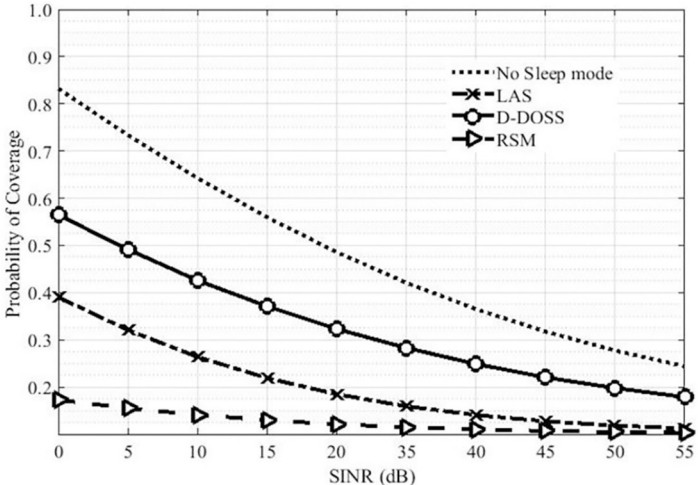

**Fig 9. Coverage probability Vs SINR.**

results that EE can be improved to a considerable level with D-DOSS. LAS is less efficient as compare to D-DOSS because of use of static/pre-defined data profile. For example, there may be a situation in which the network load is below a threshold level (from 4hrs to 6hrs) but due to static data profile BSs remain in the active state which eventually turns into a decrease in EE which can be noticed from the Fig 11. EE improvement monotonically increasing with increasing number of BS which may leads to high probabilirt of finding the "righ" BS in active mode, the superioroty of proposed D-DOSS strategy can be verifed from the plot. EE can be improved by 67% approximatly by employimg D-DOSS strategy. It can also be observed that a network with no sleep mode scheems provides the lower bound for EE. The prime reason for the improvement in EE of the network is turning off the unneccesary BSs which are transmitting power is lighly loaded regions. With D-DOSS an energy improvement of around Introducing sleep modes can improve the system performance but it is worth noting that the optimum selection of proper sleep strategy can increase the overall networks.

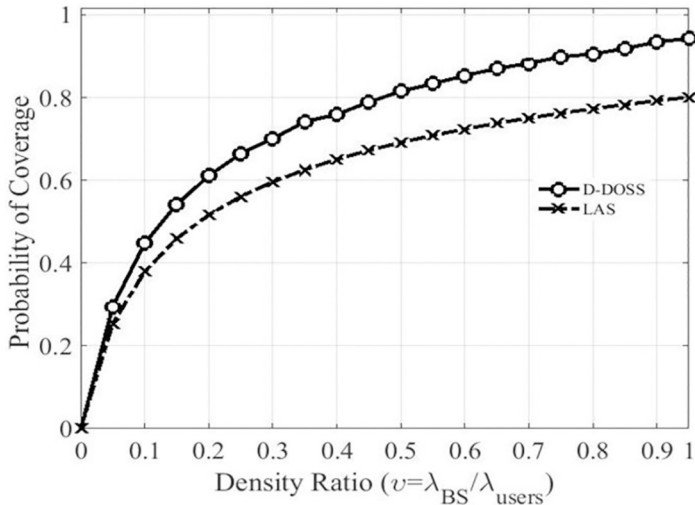

**Fig 10. Coverage probability Vs threshold ratio.**

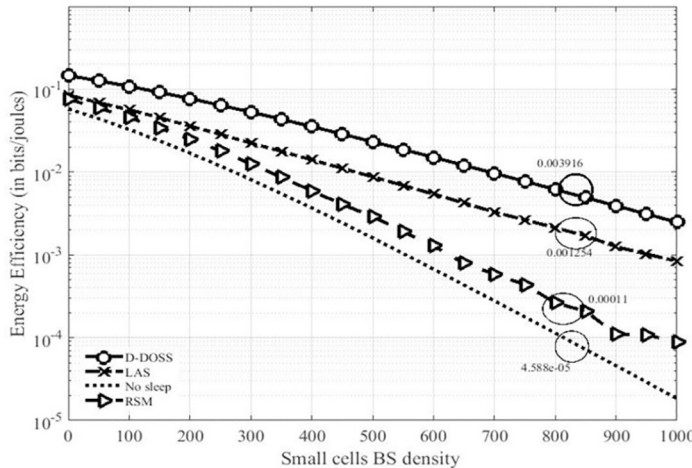

**Fig 11. Energy efficiency of HetNets.**

## 15. Performance metrics

### 15.1 Data throughput

EE sleep mode strategies have a big impact on the average achievable data rate provided by the network. Fig 12, shows that a network with no sleep mode has a nonvarying average user rate and sum rate, the reason being the constant number of active BS density throughout the network. The proposed D-DOSS approach the upper bound much faster than LAS and RSM in both average user data rate and average network data rate, because of its superior number of active BSs. Although the network with no sleep has maximized both average data rate and network data rate, but at the expense of significant energy consumption which eventually effects EE.

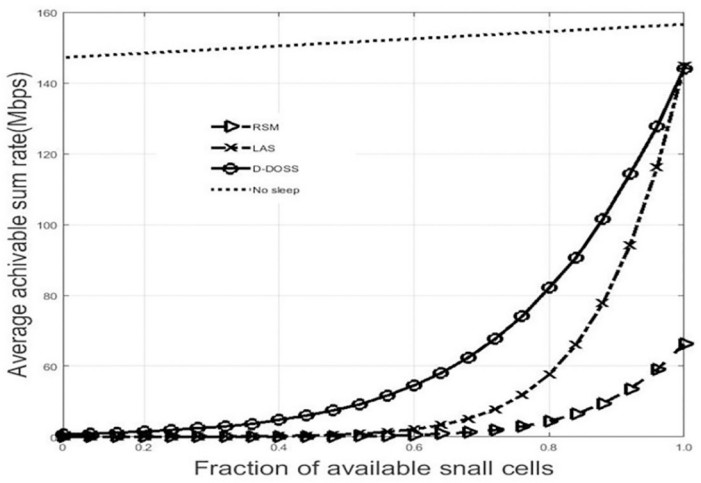

**Fig 12. Average achievable sum rate.**

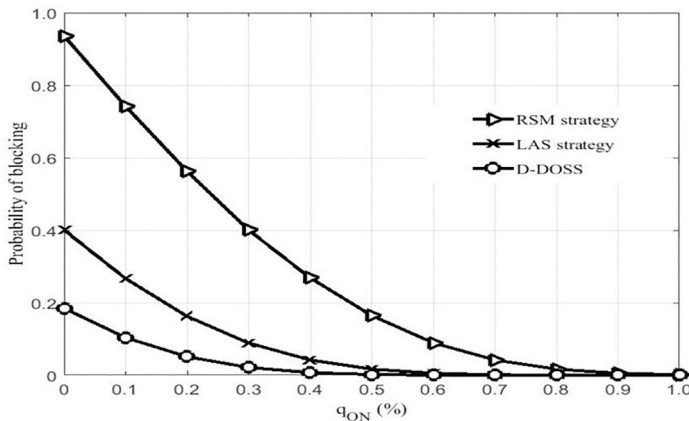

**Fig 13. Probability of blocking.**

## 15.2 Blockage probability

Fig 13, shows the average blocking probability w.r.t to an increasing number of active BSs. The proposed algorithm has a very slight impact on blocking probability which can be observed from simulated results. The reason for low blocking probability is the timely access of nearest BS.

The proposed algorithms grantees the accessibility of active BS (with less number of BSs) even when serving BS is switched off by shifting mobile users to its nearest active BS before turning off the parent BS and provide a better opportunity to other small cell BS with relatively low utility to be switched-off. When compared to other two strategies, RSM algorithm switched off the BSs w.r.t to a probability distribution which increases the chance of call blocking. However LAS will have improved performance in comparison with RSM by switching on more BSs according to the pre-define data profile. Nevertheless, the D-DOSS strategies out performs the other two scheme by reducing blocking probability considerably especially when a number of serving BS is fairly less in numbers i-e from 0 to 5.

This will have the high impact on QoS experienced by a randomly selected user especially during busiest hours. When the data traffic requirements are increasing from medium to high intensity, more BS needs to stay in an active mode which is not possible with RSM due to well define switching on and off of small cell BSs. Blocking probability is improved by using LAS algorithm but the overall probability is still high due to the un-appropriated assignment of small cell BS i-e a BS can only serve pre-define number of the user according to it data traffic profile. Hence due to the limited resources, some newly arrived mobile users in need of more active BS will be blocked which will lead towards the decreasing QoS of the entire network. This situation can be improved by using a proposed D-DOSS algorithm which will improve the desired QoS by shifting that blocked user to other BSs due to its dynamic.

## 16. Conclusion

In conclusion, this research has provided a comprehensive perspective on the challenges and opportunities in achieving energy efficiency (EE) in multi-tier Heterogeneous Networks (HetNets). A critical analysis of existing literature underscores the need for more comprehensive strategies to transform HetNets into genuinely energy-efficient networks. The study's significant contributions encompass several vital dimensions: 1. Connectivity Model: The development of a Poisson Point Process (PPP)-based connectivity model using stochastic geometry is

a foundational element for evaluating network performance, reducing congestion, and avoiding blocking. 2. D-DOSS Strategy: The introduction of the Dynamically Distributed Opportunistic Sleep Strategy (D-DOSS) presents a fresh approach to enhancing energy efficiency in HetNets. By prioritizing sleep mode activation in base stations according to specific criteria, this strategy has the potential to significantly boost real-world network energy efficiency. 3. QoS Framework: The proposed analytical framework for Quality of Service (QoS) provision, built upon the D-DOSS strategy, offers a deeper understanding of the intricate relationships among users, small cell active/sleep base stations, and macro base stations. It simplifies complex analyses and provides crucial metrics for evaluating network performance. 4. Optimization Problem: Formulating an optimization problem that accounts for coverage, energy maximization, and delay minimization constraints represents a critical step in the pursuit of EE HetNets. This structured approach aims to strike a balance between these fundamental network attributes. The research methodology outlined in this work addresses fundamental challenges in establishing EE HetNets and introduces innovative solutions to complex issues often unaddressed by existing approaches. The insights and findings of this study are poised to guide future endeavors in developing more efficient and sustainable wireless networks, benefiting both network operators and end-users. As the demand for high-speed data continues to escalate and the number of wireless devices proliferates, the need for energy-efficient HetNets becomes increasingly paramount. This research forms a cornerstone upon which forthcoming advancements in network design, operation, and management can be constructed, contributing to a more sustainable and responsive wireless communication landscape.

## 17. Limitations and future recommendations

This paper acknowledges certain assumptions made for analytical simplicity. Future research can expand the proposed D-DOSS model by considering multiple tiers, evaluating centralized schemes for potential performance improvements, and optimizing energy consumption through a centralized approach within clusters. Additionally, incorporating long-term shadowing analysis will provide a more realistic depiction of signal propagation, and exploring the impact of realistic traffic attributes on system performance beyond the assumption of full buffering in active Base Stations (BSs) will enhance the model's practical relevance.

These future research directions are crucial for advancing the current study, addressing limitations, and broadening the applicability of the proposed D-DOSS framework. By delving into these avenues, researchers can refine the model's accuracy and effectiveness in capturing the complexities of diverse operational scenarios.

## Author Contributions

**Conceptualization:** Amna Shabbir, Safdar Rizvi.

**Funding acquisition:** Muhammad Mansoor Alam, Mazliham Mohd Su'ud.

**Investigation:** Amna Shabbir.

**Methodology:** Amna Shabbir, Safdar Rizvi.

**Project administration:** Muhammad Mansoor Alam.

**Supervision:** Muhammad Mansoor Alam, Mazliham Mohd Su'ud.

**Writing – original draft:** Amna Shabbir.

**Writing – review & editing:** Muhammad Mansoor Alam, Faizan Shirazi.

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
