## [Decision Letter · Decision Letter 0]

21 Nov 2023

PONE-D-23-35635Optimizing energy efficiency in heterogeneous networks: an integrated stochastic geometry approach with novel sleep mode strategies and QoS frameworkPLOS ONE

Dear Dr. Suud,

Thank you for submitting your manuscript to PLOS ONE. After careful consideration, we feel that it has merit but does not fully meet PLOS ONE’s publication criteria as it currently stands. Therefore, we invite you to submit a revised version of the manuscript that addresses the points raised during the review process. Please submit your revised manuscript by Jan 05 2024 11:59PM. If you will need more time than this to complete your revisions, please reply to this message or contact the journal office at plosone@plos.org. Please include the following items when submitting your revised manuscript:A rebuttal letter that responds to each point raised by the academic editor and reviewer(s). You should upload this letter as a separate file labeled 'Response to Reviewers'.A marked-up copy of your manuscript that highlights changes made to the original version. You should upload this as a separate file labeled 'Revised Manuscript with Track Changes'.An unmarked version of your revised paper without tracked changes. You should upload this as a separate file labeled 'Manuscript'.

We look forward to receiving your revised manuscript.

Kind regards,

Vincent Omollo Nyangaresi, Ph.D

Academic Editor

PLOS ONE

Journal Requirements:

Reviewers' comments:

Reviewer's Responses to Questions

**Comments to the Author**

1. Is the manuscript technically sound, and do the data support the conclusions?

Reviewer #1: Partly

Reviewer #2: Yes

2. Has the statistical analysis been performed appropriately and rigorously? 

Reviewer #1: Yes

Reviewer #2: Yes

3. Have the authors made all data underlying the findings in their manuscript fully available?

Reviewer #1: Yes

Reviewer #2: Yes

4. Is the manuscript presented in an intelligible fashion and written in standard English?

Reviewer #1: Yes

Reviewer #2: Yes

5. Review Comments to the Author

Reviewer #1: Authors proposed optimizing energy consumption based on stochastic geometry with QoS framework. Here are some comments:

1- In abstract, the value achieved in improving energy consumption and its difference from the most efficient method must be mentioned.

2- In abstract, the research methodology is not well articulated and is somewhat messy.

3- In the introduction, Figure 1 does not clearly list the purpose of its existence in terms of energy efficiency over the recent years.

4- In line 70, the fundamental challenges are not mentioned

5- In the line 83, it is better to delete a (random)

6- In line 99, what are the three approaches, it should be listed and explain what it is HetNets in line 100.

7- In related works, due to the importance of both DAS and LAS, its working mechanism must be explained.

8- It is preferable to divide the related work according to common mechanisms.

9- The “Related Work” and “Introduction” sections lack of enough references. I strongly recommend that the author improve this section by adding references that support all the claims, the challenges of QoS, and motivation of the problem. The author may precisely and comprehensively point out the current issues and existing solutions. I suggest adding more related reference such as:

https://ieeexplore.ieee.org/abstract/document/9799997

https://link.springer.com/chapter/10.1007/978-981-19-1653-3_9

https://link.springer.com/chapter/10.1007/978-3-031-19523-5_6

https://ieeexplore.ieee.org/abstract/document/9768338

https://link.springer.com/chapter/10.1007/978-3-030-95987-6_1

10-In line 169, please replace oversimplified � simplified

11- In lines 174-175, “EE algorithms based on sleep mode strategies, the negative impact on QoS (174 when BSs are turned 175 into sleep mode) has been ignored in various research the publications”, What is the proof for this claim?

12-Contributions are worded superficially and do not fit with the proposed work. They must be explained in depth

13- Section 6 . Mathematical Preliminaries, Lacks references

14- The role of stochastic geometry in the work is not entirely clear

15- In line 237, “that the density of the points will remains constant throughout the Euclidean space.”, the constant of these points may affect accuracy

16- In section 7. System model, There is a bit of ambiguity and messy as to whether the proposed work is two-tier or multi-tier

17-In line 267, these conditions were not mentioned or clarified

18- In lines 284-285, this wake-up time requires clarification

19- Table 3, how were these values assigned?

20- In lin2 309, the percentages are 67% and 98%, how were they obtained, and the context of the work is here

21- The variables and factors of Equation 13 and 16 are not explained

22- In line 22, the transmission power was assumed to be constant, which may affect the validity of the results of the proposed work

23- The algorithms are good in terms of formulation, but require more clarification to explain the role of each one

were these values established?

Reviewer #2: 1- It is possible to add references to Table 1 according to the mentioned distribution

2- The research papers in Related Work are not arranged according to year of publication (from oldest to newest). In addition to the possibility of mentioning the failures of each research.

3- It is possible to discuss the results and build clear and scientific conclusions, in addition to setting a future outlook for researchers in this field

4-Standardize the format of references

6. PLOS authors have the option to publish the peer review history of their article (what does this mean?). If published, this will include your full peer review and any attached files.

Reviewer #1: No

Reviewer #2: No

---

## [Author Response · Author response to Decision Letter 0]

30 Nov 2023

Dear Mia Vanessa Recto,

I have thoroughly addressed the specific comments provided by the reviewers and the editor. The revised manuscript, incorporating these changes, is now available for your review. I appreciate the valuable feedback received and look forward to your assessment.

Best regards

---

## [Decision Letter · Decision Letter 1]

13 Dec 2023

Optimizing energy efficiency in heterogeneous networks: an integrated stochastic geometry approach with novel sleep mode strategies and QoS framework

PONE-D-23-35635R1

Dear Dr. Suud,

We’re pleased to inform you that your manuscript has been judged scientifically suitable for publication and will be formally accepted for publication once it meets all outstanding technical requirements.

Kind regards,

Ji-Hoon Yun

Academic Editor

PLOS ONE

Reviewers' comments:

Reviewer's Responses to Questions

**Comments to the Author**

1. If the authors have adequately addressed your comments raised in a previous round of review and you feel that this manuscript is now acceptable for publication, you may indicate that here to bypass the “Comments to the Author” section, enter your conflict of interest statement in the “Confidential to Editor” section, and submit your "Accept" recommendation.

Reviewer #1: All comments have been addressed

Reviewer #2: All comments have been addressed

2. Is the manuscript technically sound, and do the data support the conclusions?

Reviewer #1: Yes

Reviewer #2: Yes

3. Has the statistical analysis been performed appropriately and rigorously? 

Reviewer #1: Yes

Reviewer #2: Yes

4. Have the authors made all data underlying the findings in their manuscript fully available?

Reviewer #1: Yes

Reviewer #2: Yes

5. Is the manuscript presented in an intelligible fashion and written in standard English?

Reviewer #1: Yes

Reviewer #2: Yes

6. Review Comments to the Author

Reviewer #1: The authors made most of the revisions assigned to them, and the quality of the article now qualifies for acceptance. I recommend accepting the article

Reviewer #2: (No Response)

7. PLOS authors have the option to publish the peer review history of their article (what does this mean?). If published, this will include your full peer review and any attached files.

Reviewer #1: No

Reviewer #2: No

---

## [Editor Report · Acceptance letter]

8 Jan 2024

PONE-D-23-35635R1 

PLOS ONE

Dear Dr. Su’ud, 

I'm pleased to inform you that your manuscript has been deemed suitable for publication in PLOS ONE. Congratulations! Your manuscript is now being handed over to our production team.

Kind regards, 

on behalf of

Dr. Ji-Hoon Yun 

Academic Editor

PLOS ONE